# The Impact of Work Engagement on Future Occupational Rankings, Wages, Unemployment, and Disability Pensions—A Register-Based Study of a Representative Sample of Finnish Employees

**Jari J. Hakanen [1,*], Petri Rouvinen [2] and Ilkka Ylhäinen [2]**

1   Finnish Institute of Occupational Health, FI-00032 Työterveyslaitos, Finland
2   ETLA, The Research Institute of the Finnish Economy, Arkadiankatu 23 B, 00100 Helsinki, Finland; petri.rouvinen@avance.com (P.R.); ilkka.ylhainen@etla.fi (I.Y.)
*   Correspondence: jari.hakanen@ttl.fi; Tel.: +358-40-5625433

**Abstract:** Despite ample interest in the potential consequences of work engagement over the last two decades, the question of whether work engagement predicts proximal and more distal career-related outcomes has gained surprisingly little attention. Using Conservation of Resources (COR) theory and a sustainable careers framework, the aim of this study was to investigate whether work engagement predicts register-based outcomes of wages, moves in occupational rankings, unemployment, and disability pensions. We used nationally representative survey data ($n = 4876$; response rate 68.7%) on Finnish employees derived from the Quality of Work Life Survey (QWLS) and matched respondent data to the Finnish Longitudinal Employer–Employee Data (FLEED), which covered the period 2013–2015. We utilized ordinary least squares (OLS) and instrumental variable (IV) estimations to strengthen causality in the analyses. Even after controlling for outcomes at baseline and several covariates, e.g., health, we found that work engagement positively predicted future wages and the probability of rising in occupational rankings, and negatively predicted future unemployment and disability pensions. This study extends the scope of the possible benefits of work engagement for employees, organizations, and society at large and contributes to career research by indicating the importance of work engagement for objectively measured indicators of sustainable careers.

**Keywords:** work engagement; sustainable careers; sustainability; career; longitudinal; instrumental variable; wages; disability pension; occupational ranking; unemployment

## 1. Introduction

Over the last two decades, work (or employee) engagement—a positive, affective-motivational, state of well-being at work—has become a widely studied topic in organizational psychology and behavior [1], human resource management [2], occupational health psychology [3], epidemiology [4], and vocational behavior [5]. As well as in academic research, employee engagement has gained wide attention in business among practitioners and consultancy firms aiming to boost company performance and productivity [6]. Existing research has indicated that feeling engaged at work is important as it seems to relate to several positive employee and organizational outcomes [7,8]. Despite being a highly popular topic of research, it is surprising that one of the fields and possible consequences of work engagement has been largely ignored, namely, its relationship with various career-related labor market outcomes. These outcomes are, however, important for employees, organizations, and society at large. Finding that work engagement would simultaneously predict better occupational success through raising both wages and occupational rankings and lowering the likelihood of unemployment and disability pensions would lend robust support to the idea that work engagement is a sustainably positive state of employee well-being.

Research on the possible consequences of work engagement tend to revolve around rather proximal outcomes such as job performance and organizational commitment, which are also posited as final outcomes of the motivational process in the job demands–resources (JD-R) model [1,9]. Both theoretically and practically, it would be valuable to gain additional understanding of whether work engagement also has more distal, career-related consequences that are beneficial and important for employees, organizations, and labor markets or whether the benefits of work engagement are short lived. In addition, although a growing number of studies have used longitudinal study designs, most of the research on the outcomes or consequences of work engagement are also proximal in the sense that they are based on cross-sectional, self-reported data that cannot properly address causality [10]. Moreover, the data of most studies on work engagement have included either one or very few occupations or organizations, and studies that use samples that are representative of the general working population are sparse. Statistically, using heterogeneous samples covering a variety of jobs has been recommended, as the restriction of including only one or a very limited number of jobs may reduce the observed correlations between the study variables and produce lower estimates of population correlations [11].

In this study, we integrated work engagement research with the career literature, in particular the recently introduced sustainable careers concept [12] and Conservation of Resources (COR) theory [13]. We contribute to research on work engagement and the JD-R model by expanding the scope of the potential outcomes of work engagement to career outcomes based on the assumption that work engagement implies a resource reservoir that can lead to future resource gains [13,14]. We also contribute to career research by testing whether work engagement—which we consider a subjective indicator of sustainable careers—predicts other objectively measured indicators of sustainable careers related to productivity and health [15] and whether work engagement also protects from negative career shocks [16,17]. More specifically, the present study used a large representative sample of the Finnish working population and, over a two-year follow-up period, examined whether work engagement predicted objectively measured, register-based, career-related outcomes, i.e., positively predicted wages and moves in occupational rankings and negatively predicted unemployment and disability pensions. To strengthen the robustness of our findings and the causality of the tested relationships, we employed both the ordinary least squares (OLS) and the instrumental variable (IV or two-stage least squares, 2SLS regression) approaches. When randomized control trials are not an option, the IV approach is one of the strongest types of quasi-experimental designs [18], as it aims to take into account the endogeneity of the predictor—in the present study, work engagement—i.e., the correlation between the regressor and the error term. Endogeneity may be present in observational data in three instances: in the errors in the measures used, in the omitted but relevant variables, and in simultaneous reciprocal causality [18]. The IV approach is common in econometrics [19] but practically never utilized in work and organizational psychology, despite its use being strongly advocated in leadership and management research [18] as well as in social sciences more generally [20].

## 1.1. Work Engagement and Its Outcomes

There are several conceptualizations of employee engagement [21], the most popular in research being work engagement. Work engagement is defined as "a positive, fulfilling state of mind that is characterized by vigor, dedication, and absorption" [22] (p. 74) and is operationalized by the Utrecht Work Engagement Scale (UWES). Vigor refers to high levels of energy and mental resilience while working, the willingness to invest effort in one's work, and persistence in the face of difficulties. Dedication in turn refers to a sense of significance, enthusiasm, inspiration, pride, and challenge. The third defining characteristic of engagement is absorption, which is characterized by being fully concentrated on and happily engrossed in one's work, a sense that time passes quickly and even possible difficulty to detach oneself from one's work.

According to the JD-R model [1,9], work engagement is a function of various job (and personal) resources that build both affective well-being and energetic motivation; that is, engagement at work. Drawing on COR theory [13], as a positive high arousal state, work engagement can be considered a resource reservoir or a surplus resource (energy) that results from possessing sufficient job (e.g., autonomy, skill variety, feedback, support) and personal resources (e.g., professional efficacy, optimism, sense of coherence) to exceed the demands of one's job [23,24].

Much research has shown that engaged employees indeed seem to possess surplus resources that can lead to further resource gains, as work (or employee) engagement has been found to be positively linked to many employee and organizational outcomes [10,25], such as job performance [26], business-unit productivity measures [6], customer loyalty [27], (fewer) sickness absences [28], and work ability [29]. In addition, work engagement has been associated with better health and general well-being; for instance, healthy cardiac autonomic activity [30], mental health in terms of low levels of depression and high levels of life satisfaction [31], better cortisol suppression in response to dexamethasone [32], lower systolic blood pressure [33], less future burnout [34] and less negative and more positive spillover from work to family life [24].

However, it has also been suggested that work engagement may have a dark side [35]. Having surplus resources could lead engaged employees to invest too much of their resources in work, which could lead to workaholism [35] or to the depletion of energetic resources and burnout [36]. A study by Halbesleben et al. [23] found support for the downside of work engagement, as work engagement predicted more work-family conflict over time due to investing surplus resources in organizational citizenship behaviors. In addition, Kinman & Wray [37] found in their cross-sectional study that work engagement positively associated with presenteeism; that is, working despite feeling sufficiently unwell to justify taking sick leave. Presenteeism is a well-known risk factor for productivity losses and future health problems [38]. These few studies, in contrast to many others, suggest that work engagement could also act as a double-edged sword: Heavy resource investments at work could have negative consequences that might extend to affecting future sustainable career outcomes. Therefore, it is also important to extend the research on work engagement to its impacts on career-related outcomes in order to test whether work engagement is a sustainably healthy and productive state of employee well-being.

### 1.2. Work Engagement and Sustainable Careers

Despite ample research on engagement at work, surprisingly little evidence exists on whether it also leads to better career-related outcomes. For instance, a recent review of research on work engagement and careers [39] mentioned only three subjective career-related outcomes—namely, career commitment, satisfaction, and turnover intentions. Similarly, a narrative synthesis on employee engagement only mentioned turnover intentions as a career-related outcome of engagement [10]. To fill this notable gap, the present study focused on the temporal relationships between work engagement and four objectively measured career outcomes: increases in wages, moves in occupational rankings, unemployment, and disability pensions. Theoretically, we integrated COR theory [13] and the career literature, particularly related to sustainable careers, to test whether work engagement leads to better career outcomes over time.

Career research has long focused on the distinction between subjective and objective career success [40]. Career success has been defined as the accumulated positive work and psychological outcomes resulting from one's work experiences [41]. Recently, based on the sustainable careers concept [42], a model of sustainable careers was introduced [15]. The model highlights a long-term perspective and mutually beneficial consequences for the person and for their surrounding context. It also defines health (mental and physical), happiness (engagement, satisfaction), and productivity (performance, career potential) as three proxy indicators of sustainable careers. The authors do not recommend investigating

these indicators separately, as they are interrelated and together in tandem characterize the sustainability of careers.

In the present study, on the basis of what is known of the positive work- and more general nonwork-related outcomes of work engagement, we consider work engagement to indicate the subjective happiness dimension of sustainable careers. As regards the productivity dimension, we investigated increases in wages and moves in occupational ranking. As a health indicator, we included an ultimate work-related ill-health indicator—namely, disability pensions that follow from various chronic mental and physical health problems. Whereas work engagement can be labelled an indicator of subjective career success, these outcomes can also be called objective career success indicators, which have been clearly less often studied in career research than subjective indicators [43].

The sustainable career model also underlines the importance of considering person, context, and time together, as they often interact. One example is various career shocks [15], which can be described as disruptive and extraordinary events that are, at least to some degree, caused by factors outside the individual's control and that trigger a deliberate thought process concerning one's career [16]. In this study, we focused on one typical negative career shock—namely, unemployment. Although unemployment is not synonymous with job loss, an obvious career shock, the latter typically leads to the former, as for instance only 1% of the unemployed in Finland are estimated to be voluntarily unemployed [44]. In addition to its manifest functions (financial situation), employment has several latent functions; for example, social contacts, time structure, status, activity, collective purpose, and the psychological need of competence [45]. A lack of these functions has been found to mediate the relationship between unemployment and distress, and re-employment in turn has predicted gains in these functions, which in turn have led to reduction in distress [45]. Therefore, unemployment can be considered a specific kind of career shock.

COR theory was originally a stress theory and accordingly has been generally applied in numerous studies of occupational health and well-being. Interestingly, it has also been applied in several career-related studies. For instance, in their review, Spurk et al. [40] compared the antecedents and outcomes of objective and subjective career success using COR theory. In addition, in their meta-analysis, Ng and Feldman [46], using COR theory, found support for their hypothesis that different career hurdles impede employees' resource acquisition capacity and are therefore negatively related to salary attainment. Barthauer et al. [47] used COR theory's assumptions to investigate burnout and unsustainable careers. Moreover, Kelly et al., [48] employed the COR framework to study the effects of time spent on leisure to career-related resources of self-efficacy and resilience. Finally, De Vos et al. [15] adapted COR theory and its principles of resource investments and resource conservation when developing their conceptual model of sustainable careers.

According to COR theory's main tenet, people strive to obtain, protect, and foster the things that they value [13]. These valued entities Hobfoll [13] labelled resources, which according to the theory can be material (e.g., objects, money), conditions (e.g., supportive work climate), personal characteristics (e.g., optimism), or different forms of energy (e.g., motivation, time). COR theory also postulates that those with greater resources are less vulnerable to various resource losses and more capable of new resource gains. Thus, those with resource reservoirs have a tendency to accumulate and enrich resources over time, i.e., a tendency toward resource caravans indicated by different positive consequences [13]. Therefore, as suggested by COR theory, the consequences of the motivational process driven by job resources and work engagement in the JD-R model can be expected to go beyond immediate outcomes such as performance and commitment and also cover career-related outcomes. We can expect work engagement to lead to various positive career outcomes, such as increases in wages and positive changes in occupational rankings and protection from resource losses such as disability pensions due to health problems, and negative career shocks such as unemployment.

*1.3. The Present Study*

In this study we tested whether work engagement would lead to better objectively measured indicators of sustainable careers. To strengthen the assumption of the causality of the relationships between work engagement and career-related outcomes, we employed both the most often used OLS and the rarely used IV estimation approach [18,49]. Using these two analytic procedures, we aimed to indicate that work engagement and outcomes are not only related over time, but that work engagement indeed predicts the outcomes.

Already in 2011, in the special issue of the *European Journal of Work and Organizational Psychology* on work engagement, George [50] (p. 55) underlined that "the work engagement literature is largely silent on the fact that extrinsic work outcomes are important to employees". As existing work engagement theory and research have one-sidedly underlined the intrinsic rewards for the employee of being engaged, it has largely ignored the fact that engaged employees contribute to the success of their companies. Thus, from the distributive justice and social exchange perspective, it would seem fair that they also receive more in return, in the form of, for example, increased wages and positive occupational moves.

As work engagement comprises surplus reservoirs, engaged employees are ready to invest these resources back into work, go the extra mile, and perform better and more successfully in their jobs [7]. Indeed, several studies have linked work engagement to better performance and productivity, as mentioned earlier. Engaged employees are also likely to behave proactively by, for instance, showing pursuit of learning or personal initiative [51]. By showing personal initiative, employees are likely to become even more engaged in the future [34]. Therefore, they are also more likely to be promoted to better jobs and earn a better salary.

Work engagement is also likely to be related to better employability; that is, competencies and the necessary ability to retain one's job [52]. Thereby, engaged employees may also be less likely to become unemployed. Although losing one's job is usually a negative career shock that is not under the individual's control, work engagement results from possessing job-related (e.g., skill discretion, support, appreciation) and personal (self-efficacy, optimism, sense of coherence) resources, which may support one's employability and ability to keep one's job also during organizational changes and times of downsizing. In addition, work engagement is also reciprocally related to valuable resources—for instance, job and personal resources [53]—and engaged employees are also more likely to proactively craft themselves better jobs in the long term [34]. Job crafting, in turn, can act as an employee strategy to respond and adapt to organizational change [54]. Furthermore, work engagement is also associated with better work ability [29], which may protect against losing one's job. For these reasons, work engagement is likely to lead to further valuable resources that can support remaining employed even during work life turbulence.

The JD-R model has considered job demands and burnout to be the main predictors of health and ill health, but several studies have indicated that work engagement is associated with better physical and mental health [30–32,55]. In addition, work engagement has predicted better self-rated work ability [29], which is a strong (negative) predictor of disability pensions [56]. As engaged employees seem to have surplus resources that turn into better health resources, they are also less likely to end up on disability pensions.

Based on previous theorizing and existing research, we formulated the following hypotheses:

**Hypothesis 1 (H1).** *Work engagement predicts increased wages.*

**Hypothesis 2 (H2).** *Work engagement predicts positive occupational moves.*

**Hypothesis 3 (H3).** *Work engagement negatively predicts future unemployment.*

**Hypothesis 4 (H4).** *Work engagement predicts a lower likelihood of future disability pensions.*

## 2. Materials and Methods

### 2.1. Sample

The dataset used in this study was constructed from data provided by the research laboratory of Statistics Finland. The baseline data consisted of the Finnish Longitudinal Employer–Employee Data (FLEED), analyzed over the period 2013–2015. These register-based individual-level total data cover the population of individuals aged 15 to 70 who lived in Finland during the study period. Using these data as a starting point, we matched the Quality of Work Life Survey 2013 (QWLS), conducted by Statistics Finland, to the data, based on an encrypted personal identifier. We collected the QWLS sample from Statistics Finland's Labor Force Survey, which covered participants aged 15–64 who were identified as employees (working at least 10 h per week during the time of the interview). The QWLS is based on 4876 interviews. We also supplemented the dataset with additional FLEED-sourced employment and pension data that included further information on the jobs and pensions of the individuals (e.g., starting dates, pension types).

In Finland, women exceeded the number of men among wage earners during the data collection at baseline. The respondents were wage earners representing a multitude of industries, such as agriculture, forestry, fishing, manufacturing, transport, storage, communication, public administration, social and health care, education, construction, and other service sectors. Particularly, the number of employees working in industry decreased from 29% in 1977 to 14% in 2013. The participants' mean age was 43.6 years (SD = 11.84), 53% were women, 54% were married, and 2% were of foreign nationality. Average job tenure was six years (SD = 8.58). The median total earned annual income was about EUR 35,000. The average number of unemployment months was 0.62. We estimated wage regressions as well as the regressions focusing on occupational moves for all individuals who were defined as wage earners at $t + 2$ (2015, i.e., two years post-baseline) and for whom we observed non-zero wage outcomes in the data. For unemployment and disability pensions, we utilized an extended sample that also included those who were non-wage earners during the final period.

### 2.2. Measures

Work engagement. Work engagement was assessed using three items from the Utrecht Work Engagement Scale (UWES) [22]: "At my job, I feel strong and vigorous" (vigor), "I am enthusiastic about my work" (dedication), and "I feel happy when I am working intensely" (absorption), which were included in the survey. A similar three-item version of the UWES was recently validated using national samples from five countries and was shown to be as psychometrically sound as the often used nine-item version (UWES-3) [57]. The present survey used two items that differ from the UWES-3. However, using the first author's database, which includes 36,507 employees from Finland, both the slightly different three-item UWES versions correlated at 0.94. In addition, the three items used in the present study correlated at 0.97 with the most often used UWES-9 in the same database. Cronbach's alpha was = 0.81. The items were rated on a 4-point scale, ranging from 1 (fully agree) to 4 (fully disagree). We reversed the scale so that the higher value referred to higher work engagement.

Outcomes. ln(Wage) was a natural logarithm of total earned wage and salary income at baseline ($t$) and two years later at $t + 2$. We also analyzed increases and decreases in the occupational levels between $t$ and $t + 2$ (two years post-baseline) using occupational move upward dummy (Occupation up) and a downward dummy Occupation down) dummies. The occupational levels were defined using the hierarchy of the International Labour Organization's (ILO) International Standard Classification of Occupations (ISCO) and included main categories of managers, professionals, technicians and associate professionals, and workers. We measured Unemployment in terms of the unemployment months experienced during a year. Finally, Disability pension was a dummy variable that took a value of one for individuals who obtained disability pensions during a year, and otherwise zero.

Instrumental variable. We used a lottery variable as the IV for work engagement. Specifically, the survey asked the following question: "If you received so much money from, for example, the lottery or inheritance that you could live comfortably without working, what would you do: (1) stop working altogether, (2) work every now and then, (3) attempt to significantly shorten working time, (4) or continue working as currently?" As the instrument, we constructed a dummy which took a value of one if the respondent answered that they would continue working as currently and zero otherwise. It seems plausible that the above instrument would satisfy the requirements of relevance and validity. First, the instrument should be relevant, because it captures individuals who are likely to be "very enthusiastic" and highly intrinsically motivated for their work, as they would continue working as currently, regardless of their pay. Second, it seems reasonable to expect that the instrument would be valid because, by definition, the above measure should not affect wages other than through work engagement. The same holds for other outcomes consisting of occupational moves, unemployment and disability pensions; the instrument, by definition, would capture individuals who—controlling for their health—do not deliberately plan to slow down work or stop working altogether, suggesting that the lottery instrument is exogenous. In the present sample, 17% of the participants reported that they would continue working as currently even if they won the lottery or received inheritance.

Control variables. The individual-level variables were based on the register data obtained from FLEED. The only exceptions were related to weekly working hours and two indicators of health status, which we obtained from QWLS. The individual-level control variables included age (Age) and its square, dummies for gender (Female), marital status (Married), education (High school, Bachelor, Master, and Licentiate/Doctor), foreign nationality (Foreign), weekly working hours (Hours worked), number of children under 18 (Number of children), and years of experience in current job (Tenure). If the individual had multiple jobs, we used the one in which they had been the longest. Furthermore, we controlled for the health of individuals: Chronic disease was an indicator that the individuals had a chronic disease (e.g., cardiovascular disease, lung disease, musculoskeletal disorder, digestive system disease, or other chronic disease). Sick leave was an indicator that the individuals had at least one long sickness absence (i.e., lasting either 4–9 days or more than ten days) in the last twelve months. The control variables also included two-digit industry, occupation, and sub-region dummies. We chose these background variables because they have been found to correlate with work engagement [58]. In the analysis of unemployment, we also controlled for previous unemployment, measured in terms of unemployment months experienced during the calendar year preceding initial period t. This measure controlled for individuals who had a higher tendency to be exposed to unemployment than other individuals. In the analyses of disability pensions, we also controlled for the impact of baseline disability pension status, as some of those who were on disability pension might still have been working on a part-time basis, allowing them to respond to the survey.

### 2.3. Statistical Analyses

The estimated model for wage and salary income was defined as follows:

$$\underbrace{y_{it+2}}_{\ln(Wage_{2015})} = \propto + \underbrace{x'_{it}\beta}_{\substack{Controls_{2013} \\ (incl.\ initial\ Wage)}} + \underbrace{\gamma z_{it}}_{Work\ engagement_{2013}} + \varepsilon_{it} \qquad (1)$$

where the dependent variable was measured at $t + 2$ (2015) and the independent variables were measured at $t$ (2013). We tested similar models for other outcomes.

The baseline specification does not account for the potential endogeneity of work engagement. Such an endogeneity problem could arise due to reversed causality. For example, work engagement may have an impact on wages, but it could also be the other way around, i.e., wages may affect work engagement. It is also possible that some unobserved

factors that correlated with work engagement were omitted. To control for these issues, in addition to using OLS regression analyses, we also applied an IV estimation, using two-staged least squares (2SLS).

Because we treated work engagement as endogenous, we needed an IV that would correlate with work engagement but not with the error term. The IV must not directly relate to the dependent variable—only through its effect on work engagement. The estimation of the IV model using 2SLS can be illustrated as follows: In the first-stage regression, the endogenous variable was regressed against the IV and the rest of the exogenous variables. In the second-stage regression, predicted values from the first-stage regression were plugged into the regression model, Equation (1), shown above in place of the endogenous variable. Using this statistical procedure, we were able to address the problem of endogeneity and draw conclusions regarding the effects of work engagement on career-related outcomes in causal terms.

To test that the chosen instrument—here the lottery question—was not a weak predictor of (endogenous variable) work engagement, we used F statistics. The F statistic is the most robust and conservative test of an instrument's strength [49]. As a rule of thumb, values above 10 indicate acceptable instruments, not weak ones [59]. As reported in the Results, our instrument clearly fulfilled this criterion in all the IV analyses.

### 3. Results

*3.1. Descriptive Statistics*

Table 1 shows the means, standard deviations, and correlations of the study variables.

*3.2. Results of OLS Analyses*

During the two-year follow-up period, we found support for Hypothesis 1, as the OLS regression analyses results indicated that after controlling for baseline wage level and many other factors, work engagement predicted increases in future wages according to the OLS ($\beta = 0.08$, $p < 0.001$). Table 2 also shows that gender (men), curvilinear age (being middle-aged), and education (Master) were positively related to future wages. In contrast, long sick leaves were negatively related to future wages.

Next, using OLS, we found no relationship between occupational moves upward or downward and work engagement. Thus, Hypothesis 2 was not supported. As could be expected, education was positively related with the probability of rising in occupational ranks in the future.

As can also be seen from Table 2, work engagement was negatively related to future unemployment ($\beta = -0.17$, $p < 0.01$), even after the unemployment periods during the previous year and health indicators of individuals were controlled for, thus supporting Hypothesis 3. In addition, being married was negatively and sick leaves were positively related to future unemployment.

Finally, the OLS estimates indicated a negative and significant relation between work engagement and future disability pension ($\beta = -0.01$, $p < 0.05$). Thus, in line with Hypothesis 4, work engagement reduced the probability of future disability pensions, even after we controlled for the impact of disability pension status and health (sick leaves and chronic diseases) at baseline, which were all positively related to future disability pension. Gender (men) was also significantly related to disability pensions.

**Table 1.** Means (M), standard deviations (SD), and correlations between study variables.

| Variable | M | SD | n | 1. | 2. | 3. | 4. | 5. | 6. | 7. | 8. | 9. | 10. | 11. | 12. | 13. | 14. | 15. | 16. | 17. | 18. | 19. | 20. | 21. | 22. | 23. |
|---|---|---|---|---|---|---|---|---|---|---|---|---|---|---|---|---|---|---|---|---|---|---|---|---|---|---|
| 1. Work engagement | 3.26 | 0.54 | 4774 | | | | | | | | | | | | | | | | | | | | | | | |
| 2. Lottery win | 0.17 | 0.37 | 4774 | 0.17 *** | | | | | | | | | | | | | | | | | | | | | | |
| 3. ln(Wage) $_{t+2}$ | 10.46 | 0.61 | 4268 | 0.12 *** | 0.08 *** | | | | | | | | | | | | | | | | | | | | | |
| 4. Occupation up $_{t+2}$ | 0.09 | 0.28 | 4268 | 0.02 | 0.04 ** | 0.05 ** | | | | | | | | | | | | | | | | | | | | |
| 5. Occupation down $_{t+2}$ | 0.06 | 0.24 | 4268 | 0.01 | 0.03 | 0.04 ** | −0.08 *** | | | | | | | | | | | | | | | | | | | |
| 6. Unemployment $_{t+2}$ | 0.62 | 2.18 | 4774 | −0.06 *** | −0.03 | −0.24 *** | −0.01 | 0.08 *** | | | | | | | | | | | | | | | | | | |
| 7. Disability pension $_{t+2}$ | 0.02 | 0.14 | 4774 | −0.04 *** | −0.03 | −0.19 *** | 0.01 | −0.02 | −0.01 | | | | | | | | | | | | | | | | | |
| 8. ln(Wage) | 10.45 | 0.55 | 4268 | 0.07 *** | 0.06 *** | 0.69 *** | 0.01 | 0.07 *** | −0.15 *** | −0.10 *** | | | | | | | | | | | | | | | | |
| 9. Unemployment | 0.30 | 1.20 | 4774 | 0.01 | 0.00 | −0.14 *** | 0.02 | −0.02 | 0.35 *** | 0.00 | −0.26 *** | | | | | | | | | | | | | | | |
| 10. Disability pension | 0.01 | 0.12 | 4774 | 0.00 | 0.01 | −0.14 *** | −0.01 | 0.00 | 0.00 | 0.58 *** | −0.18 *** | −0.02 | | | | | | | | | | | | | |
| 11. Age | 43.62 | 11.84 | 4774 | 0.00 | −0.01 | 0.14 *** | −0.08 *** | 0.03 * | 0.02 | 0.12 *** | 0.25 *** | −0.06 *** | 0.12 *** | | | | | | | | | | | | |
| 12. Female | 0.53 | 0.50 | 4774 | 0.05 *** | −0.08 *** | −0.23 *** | 0.00 | -0.03 * | −0.06 *** | 0.00 | −0.20 *** | −0.05 *** | 0.04 ** | 0.07 *** | | | | | | | | | | | |
| 13. Married | 0.54 | 0.50 | 4774 | 0.04 ** | 0.01 | 0.15 *** | −0.03 * | 0.03 * | −0.06 *** | −0.01 | 0.19 *** | −0.08 *** | 0.02 | 0.30 *** | −0.02 | | | | | | | | | | |
| 14. High school | 0.48 | 0.50 | 4774 | 0.05 *** | 0.04 * | 0.18 *** | 0.10 *** | 0.08 *** | −0.08 *** | −0.06 *** | 0.19 *** | −0.08 *** | −0.05 *** | −0.16 *** | 0.17 *** | 0.01 | | | | | | | | | |
| 15. Bachelor | 0.15 | 0.36 | 4774 | 0.02 | 0.00 | 0.04 * | 0.09 *** | 0.06 *** | −0.04 * | −0.03 * | 0.04 * | −0.02 | −0.03 * | −0.16 *** | 0.05 ** | 0.00 | 0.25 *** | | | | | | | |
| 16. Master | 0.16 | 0.37 | 4774 | 0.07 *** | 0.08 *** | 0.26 *** | 0.06 *** | 0.05 ** | −0.02 | −0.04 ** | 0.27 *** | −0.03 * | −0.04 ** | −0.03 * | 0.03 * | 0.08 *** | 0.41 *** | −0.18 *** | | | | | | |
| 17. Lic/Doc | 0.02 | 0.14 | 4774 | 0.03 * | 0.05 *** | 0.14 *** | −0.01 | −0.01 | −0.01 | −0.02 | 0.17 *** | −0.02 | −0.02 | 0.04 ** | −0.03 | 0.02 | 0.13 *** | −0.06 *** | −0.06 *** | | | | | |
| 18. Number of children | 0.69 | 1.07 | 4774 | 0.02 | −0.02 | 0.10 *** | 0.02 | −0.02 | −0.05 *** | −0.05 *** | 0.06 *** | −0.04 ** | −0.04 ** | −0.18 *** | −0.04 ** | 0.28 *** | 0.07 *** | 0.05 ** | 0.09 *** | 0.00 | | | |
| 19. Tenure | 5.99 | 8.58 | 4774 | −0.05 ** | 0.00 | 0.14 *** | −0.04 ** | 0.03 | −0.02 | −0.01 | 0.20 *** | −0.10 *** | −0.01 | 0.33 *** | −0.11 *** | 0.09 *** | −0.12 *** | −0.07 *** | −0.08 *** | −0.03 * | −0.09 *** | | |
| 20. Hours worked | 37.21 | 6.68 | 4774 | 0.01 | −0.01 | 0.33 *** | −0.03 | 0.02 | −0.03 | −0.15 *** | 0.44 *** | −0.04 ** | −0.27 *** | 0.06 *** | −0.20 *** | 0.07 *** | −0.04 ** | 0.01 | 0.01 | 0.06 *** | 0.06 *** | 0.11 *** | |
| 21. Foreign | 0.02 | 0.12 | 4774 | 0.03 * | 0.04 ** | −0.03 | 0.00 | 0.00 | 0.01 | −0.02 | −0.02 | 0.04 ** | −0.02 | −0.03 | 0.00 | −0.02 | −0.10*** | −0.02 | 0.00 | 0.01 | 0.01 | −0.04 * | 0.01 | |
| 22. Chronic disease | 0.37 | 0.48 | 4774 | −0.07 *** | −0.04 ** | −0.07 *** | −0.02 | −0.01 | 0.00 | 0.16 *** | −0.05 ** | 0.01 | 0.15 *** | 0.20 *** | 0.05 *** | 0.03 * | −0.10 *** | −0.06 *** | −0.06 *** | −0.02 | −0.09 *** | 0.03 * | −0.05 ** | −0.04 ** | |
| 23. Sick leave | 0.24 | 0.43 | 4774 | −0.09 *** | −0.03 * | −0.09 *** | −0.03 * | −0.04 ** | 0.02 | 0.10 *** | −0.04 ** | −0.03 * | 0.08 *** | 0.03 * | 0.05 *** | −0.04 ** | −0.12 *** | −0.04 * | −0.09 *** | −0.02 | −0.02 | 0.01 | 0.00 | 0.01 | 0.20 *** |

Note. * $p < 0.05$, ** $p < 0.01$, *** $p < 0.001$.

**Table 2.** Association between work engagement at baseline (*t*) and career-related outcomes (wages, occupational moves, unemployment, and disability pension) at *t* + 2 (two years post-baseline): OLS estimates.

| Dependent Variable | (1) ln(Wage) $_{t+2}$ | (2) Occupation Up $_{t+2}$ | (3) Occupation Down $_{t+2}$ | (4) Unemployment $_{t+2}$ | (5) Disability Pension $_{t+2}$ |
|---|---|---|---|---|---|
| Work engagement | 0.080 *** | 0.008 | −0.008 | −0.165 ** | −0.007 * |
| | (0.014) | (0.008) | (0.006) | (0.062) | (0.004) |
| ln(Wage) | 0.573 *** | | | | |
| | (0.038) | | | | |
| Unemployment | | | | 0.514 *** | |
| | | | | (0.060) | |
| Disability pension | | | | | 0.674 *** |
| | | | | | (0.055) |
| Age | 0.020 ** | 0.001 | −0.006 * | 0.019 | 0.002 * |
| | (0.006) | (0.003) | (0.003) | (0.021) | (0.001) |
| Age sq | −0.224 ** | −0.014 | 0.072 * | −0.087 | −0.016 |
| | (0.074) | (0.035) | (0.033) | (0.246) | (0.013) |
| Female | −0.138 *** | −0.014 | −0.005 | −0.005 | −0.009 * |
| | (0.018) | (0.011) | (0.010) | (0.078) | (0.004) |
| Married | −0.002 | −0.005 | 0.004 | −0.170 * | −0.003 |
| | (0.015) | (0.009) | (0.008) | (0.068) | (0.004) |
| High school | 0.000 | 0.044 *** | −0.008 | −0.123 | 0.002 |
| | (0.017) | (0.012) | (0.010) | (0.074) | (0.003) |
| Bachelor | 0.012 | 0.099 *** | −0.011 | 0.050 | 0.001 |
| | (0.021) | (0.017) | (0.014) | (0.091) | (0.004) |
| Master | 0.075 ** | 0.135 *** | −0.051 ** | 0.219 | 0.001 |
| | (0.027) | (0.019) | (0.017) | (0.113) | (0.004) |
| Lic/Doc | 0.116 | 0.116 *** | −0.108 *** | 0.291 | −0.008 |
| | (0.060) | (0.031) | (0.028) | (0.241) | (0.005) |
| Number of children | 0.015 * | 0.002 | −0.006 | −0.030 | −0.002 |
| | (0.007) | (0.005) | (0.003) | (0.028) | (0.001) |
| Tenure | 0.001 | −0.001 | −0.000 | −0.004 | −0.000 |
| | (0.001) | (0.001) | (0.001) | (0.005) | (0.000) |
| Tenure missing | 0.143 | 0.020 | 0.025 | −0.845 | 0.071 |
| | (0.145) | (0.098) | (0.076) | (0.438) | (0.039) |
| Hours worked | 0.002 | −0.001 | −0.001 | −0.008 | −0.000 |
| | (0.001) | (0.001) | (0.001) | (0.005) | (0.000) |
| Foreign | −0.074 | 0.025 | 0.012 | −0.046 | −0.007 |
| | (0.048) | (0.036) | (0.030) | (0.272) | (0.005) |
| Chronic disease | −0.021 | 0.005 | −0.001 | −0.067 | 0.014 *** |
| | (0.014) | (0.009) | (0.008) | (0.066) | (0.004) |
| Sick leave | −0.045 ** | −0.008 | −0.003 | 0.156 * | 0.012 * |
| | (0.017) | (0.010) | (0.008) | (0.075) | (0.005) |
| Observations | 4268 | 4268 | 4268 | 4774 | 4774 |
| $R^2$ | 0.553 | 0.164 | 0.151 | 0.187 | 0.377 |

Note. Two-digit industry and occupation dummies, sub-region dummies, and a constant were included but are not presented. Robust standard errors in parentheses. * $p < 0.05$, ** $p < 0.01$, *** $p < 0.001$.

### 3.3. Results of IV Analyses

To strengthen causality in the analyses, we also employed IV estimations for a more robust test of the hypotheses (Tables 3 and 4). First, similar to the OLS analyses, the IV analyses results indicated that after controlling for baseline wage level and many other factors, work engagement predicted increases in future wages ($\beta = 0.20$, $p < 0.01$). The IV estimates were notably higher than the OLS estimates, suggesting that the latter were downward biased. The robust *F* statistic (99.49) for the excluded instrument was well above the rule-of-thumb value of 10, indicating that the lottery instrument was not weak.

Second, although OLS estimates suggested no relationships, using IV estimates resulted in work engagement positively predicting occupational moves upward ($\beta = 0.16$, $p < 0.01$). The level of work engagement did not impact moves downward (Table 3). The first-stage estimates shown in Table 4 indicate that the lottery instrument had a positive coefficient and that it was highly statistically significant and in line with expectations. The robust F statistic for the excluded instrument was 100.53, which were again more than adequate.

**Table 3.** Association between work engagement at baseline ($t$) and career-related outcomes (wages, occupational moves, unemployment, and disability pension) at $t + 2$ (two years post-baseline): IV estimates.

| | (1) | (2) | (3) | (4) | (5) |
|---|---|---|---|---|---|
| **Dependent Variable** | **ln(Wage)** $_{t+2}$ | **Occupation Up** $_{t+2}$ | **Occupation Down** $_{t+2}$ | **Unemployment** $_{t+2}$ | **Disability Pension** $_{t+2}$ |
| Work engagement | 0.205 ** | 0.160 ** | −0.018 | −0.707 * | −0.047 ** |
| | (0.076) | (0.060) | (0.049) | (0.342) | (0.017) |
| ln(Wage) | 0.568 *** | | | | |
| | (0.036) | | | | |
| Unemployment | | | | 0.520 *** | |
| | | | | (0.060) | |
| Disability pension | | | | | 0.676 *** |
| | | | | | (0.053) |
| Age | 0.022 *** | 0.003 | −0.006 * | 0.013 | 0.002 |
| | (0.006) | (0.003) | (0.003) | (0.021) | (0.001) |
| Age sq | −0.247 *** | −0.039 | 0.074 * | −0.022 | −0.012 |
| | (0.074) | (0.037) | (0.034) | (0.245) | (0.013) |
| Female | −0.143 *** | −0.020 | −0.004 | 0.013 | −0.008 * |
| | (0.018) | (0.011) | (0.010) | (0.078) | (0.004) |
| Married | −0.003 | −0.007 | 0.004 | −0.158 * | −0.002 |
| | (0.015) | (0.010) | (0.008) | (0.067) | (0.004) |
| High school | 0.004 | 0.048 *** | −0.008 | −0.142 | 0.000 |
| | (0.017) | (0.012) | (0.010) | (0.074) | (0.003) |
| Bachelor | 0.010 | 0.097 *** | −0.010 | 0.060 | 0.002 |
| | (0.021) | (0.017) | (0.014) | (0.091) | (0.004) |
| Master | 0.071 ** | 0.129 *** | −0.050 ** | 0.239 * | 0.003 |
| | (0.027) | (0.019) | (0.017) | (0.114) | (0.004) |
| Lic/Doc | 0.110 | 0.106 *** | −0.107 *** | 0.334 | −0.005 |
| | (0.059) | (0.030) | (0.027) | (0.244) | (0.005) |
| Number of children | 0.014 | 0.001 | −0.006 | −0.027 | −0.002 |
| | (0.007) | (0.005) | (0.003) | (0.029) | (0.001) |
| Tenure | 0.001 | −0.001 | −0.000 | −0.004 | −0.000 |
| | (0.001) | (0.001) | (0.001) | (0.005) | (0.000) |
| Tenure missing | 0.121 | −0.003 | 0.027 | −0.778 | 0.076 |
| | (0.139) | (0.102) | (0.075) | (0.434) | (0.039) |
| Hours worked | 0.002 | −0.001 | −0.001 | −0.006 | −0.000 |
| | (0.001) | (0.001) | (0.001) | (0.005) | (0.000) |
| Foreign | −0.093 | 0.001 | 0.013 | 0.038 | −0.001 |
| | (0.049) | (0.036) | (0.029) | (0.272) | (0.006) |
| Chronic disease | −0.014 | 0.013 | −0.001 | −0.099 | 0.012 ** |
| | (0.014) | (0.010) | (0.008) | (0.069) | (0.004) |
| Sick leave | −0.036 * | 0.003 | −0.004 | 0.112 | 0.009 |
| | (0.017) | (0.011) | (0.008) | (0.079) | (0.005) |
| Observations | 4268 | 4268 | 4268 | 4774 | 4774 |
| $R^2$ | 0.542 | 0.088 | 0.150 | 0.170 | 0.356 |
| *F* | 99.495 | 100.531 | 100.531 | 119.785 | 119.746 |

Note. Two-digit industry and occupation dummies, sub-region dummies, and a constant were included but are not presented. Robust standard errors in parentheses. * $p < 0.05$, ** $p < 0.01$, *** $p < 0.001$. F stands for robust F statistic for excluded instrument.

**Table 4.** First-stage regressions for IV specifications (1)–(5) shown above in Table 4.

| Dependent Variable | (1) First-Stage Regression: Work Engagement | (2) First-Stage Regression: Work Engagement | (3) First-Stage Regression: Work Engagement | (4) First-Stage Regression: Work Engagement | (5) First-Stage Regression: Work Engagement |
|---|---|---|---|---|---|
| Lottery win | 0.209 *** | 0.210 *** | 0.210 *** | 0.217 *** | 0.217 *** |
| | (0.021) | (0.021) | (0.021) | (0.020) | (0.020) |
| ln(Wage) | 0.038 | | | | |
| | (0.022) | | | | |
| Unemployment | | | | 0.011 | |
| | | | | (0.008) | |
| Disability pension | | | | | 0.058 |
| | | | | | (0.085) |
| Age | −0.010 | −0.008 | −0.008 | −0.004 | −0.004 |
| | (0.007) | (0.007) | (0.007) | (0.006) | (0.006) |
| Age sq | 0.108 | 0.087 | 0.087 | 0.047 | 0.045 |
| | (0.077) | (0.076) | (0.076) | (0.068) | (0.068) |
| Female | 0.058 ** | 0.054 ** | 0.054 ** | 0.045 * | 0.045 * |
| | (0.020) | (0.020) | (0.020) | (0.020) | (0.020) |
| Married | 0.010 | 0.011 | 0.011 | 0.019 | 0.018 |
| | (0.019) | (0.019) | (0.019) | (0.018) | (0.018) |
| High school | −0.028 | −0.028 | −0.028 | −0.032 | −0.033 |
| | (0.021) | (0.021) | (0.021) | (0.021) | (0.021) |
| Bachelor | 0.012 | 0.014 | 0.014 | 0.017 | 0.018 |
| | (0.026) | (0.026) | (0.026) | (0.025) | (0.025) |
| Master | 0.019 | 0.026 | 0.026 | 0.027 | 0.027 |
| | (0.030) | (0.030) | (0.030) | (0.029) | (0.029) |
| Lic/Doc | 0.028 | 0.041 | 0.041 | 0.059 | 0.058 |
| | (0.060) | (0.059) | (0.059) | (0.058) | (0.058) |
| Number of children | 0.006 | 0.006 | 0.006 | 0.006 | 0.005 |
| | (0.009) | (0.009) | (0.009) | (0.009) | (0.009) |
| Tenure | −0.001 | −0.001 | −0.001 | −0.001 | −0.001 |
| | (0.001) | (0.001) | (0.001) | (0.001) | (0.001) |
| Tenure missing | 0.156 | 0.132 | 0.132 | 0.093 | 0.074 |
| | (0.132) | (0.128) | (0.128) | (0.094) | (0.094) |
| Hours worked | 0.002 | 0.003 * | 0.003 * | 0.002 | 0.003 |
| | (0.002) | (0.001) | (0.001) | (0.001) | (0.001) |
| Foreign | 0.128 | 0.128 | 0.128 | 0.123 | 0.125 * |
| | (0.067) | (0.067) | (0.067) | (0.063) | (0.063) |
| Chronic disease | −0.051 ** | −0.052 ** | −0.052 ** | −0.052 ** | −0.054 ** |
| | (0.018) | (0.018) | (0.018) | (0.017) | (0.017) |
| Sick leave | −0.071 *** | −0.070 *** | −0.070 *** | −0.081 *** | −0.083 *** |
| | (0.020) | (0.020) | (0.020) | (0.020) | (0.020) |

Note. The table presents the first-stage regression estimates for IV specifications (1)–(5) that utilize Lottery win dummy as the instrument. Two-digit industry and occupation dummies, sub-region dummies, and a constant were included but are not presented. Robust standard errors in parentheses. * $p < 0.05$, ** $p < 0.01$, *** $p < 0.001$.

Third, in line with the OLS analyses, work engagement negatively predicted future unemployment, even after unemployment during the previous year and the health indicators of individuals were controlled for. Compared to the OLS estimates, the IV estimates were even larger in absolute terms and still significant ($\beta = -0.71$, $p < 0.05$). The robust F statistic for the excluded instrument was 119.78.

Fourth and finally, Table 3 shows that work engagement negatively predicted future disability pension, and that compared to the OLS results, this effect was even larger after controlling for the endogeneity of work engagement ($\beta = -0.05$, $p < 0.01$). Thus, work engagement reduced the probability of future disability pensions, even after we controlled for the impact of disability pension status and health (sick leaves and chronic diseases) at

baseline. In this case, the robust F statistic for the excluded instrument was 119.75 and thus well above the critical value.

All in all, the IV analyses supported Hypotheses 1, 2, 3, and 4.

## 4. Discussion

This study sought to examine whether the positive affective–motivational state of work engagement predicted rarely studied, objectively measured career-related outcomes. It has recently been emphasized that being happy, healthy, and productive together form the defining characteristics of sustainable careers; therefore, they should also be studied together to indicate sustainability of one's work career [15]. We considered work engagement a subjective indicator of happiness in sustainable careers, which would simultaneously predict other types of objectively measured indicators of sustainable careers [15], such as future wages and occupational moves (indicating productivity) and disability pensions (indicating health), as well as protect from a typical negative career shock such as unemployment [16,17]. We used both OLS and IV approaches, a two-year follow-up design combining survey and register data, and a nationally representative sample of Finnish employees to study these relationships over time.

After controlling for many demographic and occupational factors, the results mainly support our hypotheses: Work engagement predicted increases in wages and a lower likelihood of prematurely retiring due to disability or encountering a career shock such as becoming unemployed. Work engagement was also related to positive occupational moves over time, according to the IV analyses.

Both salary attainment and positive moves in occupational rankings (or promotion) are career success factors [40], and in this study, we conceptualized them as indicators of productivity in the sustainable careers model [15]. By using both OLS and IV estimates, we found that work engagement predicted increases in wages. It is likely that, as engaged employees are on average better task and contextual performers [7,8], show more initiative [14], are healthier [30], and thus willing and able to work harder, they may also receive more in return and gain better wages. A previous study of public sector dentists [60] similarly found that after controlling for several work-related and demographic background variables, work engagement was positively related to procedure fees, which formed part of the dentists' wages. Interestingly, although there is a great deal of research on the association between pay level and job satisfaction, another positive indicator of employee well-being, these studies have constantly focused on pay level as an antecedent of job satisfaction but not as a potential consequence of it [61]. All in all, our findings suggest that in addition to being intrinsically motivating for employees, work engagement may also be extrinsically rewarding.

We also found partial support for the hypothesis that work engagement is positively related to occupational moves upwards, i.e., it was more likely that engaged employees' occupational status would rise over the two-year time period (e.g., an engaged professional may become a manager). This finding was supported by stricter IV analysis but not by OLS analysis. The IV approach may have ruled out the possibility of simultaneous causation, so that a higher position in occupational ranking may also predict future work engagement, as well as other sources of endogeneity, including potentially omitted other variable(s).

Taking a different perspective, de Lange et al. [62], using a two-wave panel design among 1670 Belgian employees, compared those who stayed in the same job ("stayers"), those who were promoted ("promotion makers") and those who quit and changed jobs ("external movers"). They found that low work engagement was related to changing jobs. In addition, among the promotion makers and external movers, work engagement increased over time, whereas among the stayers it remained at the same level. However, to our knowledge, the present study is the first to show that high levels of work engagement may lead to positive changes in occupational rankings, although the reverse direction is also possible.

In accordance with our expectations, even after controlling for sick leaves and chronic diseases, we also found that work engagement decreased the likelihood of ending up on disability pension—the robust indicator of the health dimension of sustainable careers in the present study. Several studies have indicated that work engagement is associated with better physical [30] and mental [31] health and work ability [29]. The mechanism linking work engagement and disability pensions is likely to relate to better health and more job and personal resources being available so that engaged employees can stay well and healthy until retirement age; that is, a process of resource accumulation, as theorized by COR theory [13]. Some previous studies also show that negative indicators of employee well-being—for example, job burnout [63]—are related to disability pensions. However, this is the first study to show that the positive state of feeling engaged at work may also protect from disability pensions.

Finally, the hypothesis that work engagement would decrease the likelihood of future unemployment was also supported. People may lose jobs due to many extrinsic conditions, for example, poor economic downturn or organizational restructurings. Unemployment can be considered a negative career shock that shapes the sustainability of one's career [15]. In our study, work engagement appeared as a career resource that may protect from such a resource threat as losing one's job. This finding does not mean that engaged employees are never dismissed. However, it suggests that engaged employees are more likely to remain employable and retain their present jobs than their less engaged colleagues.

### 4.1. Theoretical Implications

Theoretically, this study contributes to the work engagement literature by indicating that the positive consequences of work engagement may go beyond the much-studied organizational outcomes, such as job performance, and employee outcomes, such as well-being. As the evidence of the positive consequences of work engagement has begun to accumulate, the question has arisen as to whether work engagement has a dark side [35]. Thus far, it has received very limited support [23]. Our study adds to this discussion on the evidence of new types of long-term positive impacts of work engagement, suggesting generally that work engagement can be expected to boost sustainable benefits for employees and organizations, as well as for societies in general, in the form of extrinsic career success, retaining a job, and fewer premature pensions based on disabilities.

Work engagement is most often studied using the JD-R framework [64]. Even though we did not test the whole model, our study suggests that the final outcomes of the motivational and health impairment processes of the model could be expanded further to cover sustainable career indicators; for example, wages, employability and disability pensions. An important future research option would be to investigate whether the health impairment process from job demands via burnout to ill health or the motivational process from job resources via work engagement to job performance and other organizational outcomes are more relevant for sustainable careers. Obviously, burnout, the conceptual opponent of work engagement, also impacts career outcomes [34].

Through this study, in addition to fulfilling some gaps in work engagement research and drawing on COR theory, we wished to contribute to recently introduced sustainable careers research [15,42]. Global competition, technological innovations, and the need to reduce labor cost are constantly changing work life, and these changes are often followed by increased organizational turbulence, work intensification, and job insecurity. Therefore, sustainable careers—as indicated by happiness, productivity, and health—have become of utmost importance for employees and the wider social context [15]. The construct of sustainable careers is still being developed, and how to best operationalize the construct and define the relationships between the different indicators is challenging [65]. In our study, we focused on one indicator, happiness (work engagement), as a potential resource and driver for other sustainable career indicators, since work engagement has been evidenced to relate to many positive employee and organizational outcomes [10,25]. This means that these findings also support the happy–productive worker thesis [66]. Being happy and

productive at work has been considered a proxy for a sustainable career [65]. This study adds health to this thesis, as being happy in terms of work engagement not only predicted being productive but also being healthy in terms of being able to continue working without disabilities, i.e., happy workers may be both productive and healthy.

Another concept that shares clear similarities with sustainable careers is decent work. Originally proposed as a central goal of the International Labour Organization [67], this concept has four main values: freedom, equity, security, and human dignity; and six key elements: opportunities for work, freedom of choice of employment, productive work, equity, security, and dignity [68], which all fit with the sustainable careers model. More recently, decent work has been conceptualized in different ways—for example, as consisting of five components: physically and interpersonally safe working conditions, access to health care, adequate compensation, hours that allow for free time and rest, and organizational values that complement family and social values [69]. In addition, a recent study [70] used seven dimensions of quality of work (physical environment, work intensity, working time quality, social environment, skills and discretion, prospects, and earnings) to illustrate decent working conditions, which were also associated with work engagement. These conceptualizations of decent work can be considered determinants of sustainable careers. Evidently, future research could benefit from building bridges between these two umbrella concepts.

Methodologically, this study aimed to contribute to occupational psychological research by introducing the IV approach for testing study models. Antonakis and his colleagues [18] call IV methods the workhorse of econometrics and "almost too good to be true", as they are a cure for endogeneity resulting from omitted variables, measurement error, simultaneity, and common method bias. In general, IV estimates can be sensitive to instruments and may capture the local average treatment effect for the subpopulation affected by the instrument [19,71]. Ideally, the instrument would be objectively measured. However, our instrument, based on the lottery and inheritance question, was strongly associated with work engagement, met the criterion for a valid IV, and was thus most plausibly exogenous. Indeed, lotteries, gifts, and inheritances have long been utilized as sources of exogenous variation for studying entrepreneurial behavior, for example, as such natural experiments closely resemble idealized laboratory experiments [72,73]. Our OLS and IV estimates also appeared to be well in line with each other.

### 4.2. Practical Implications

From a practical perspective, the present study suggests that the benefits of work engagement may extend beyond those already identified in the literature. Organizations have several possibilities to increase work engagement and thereby promote sustainable careers. A recent systematic review of work engagement interventions [74] indicated that various interventions had targeted personal resource building, job resource building, leadership building, or health promotion. According to the review, the most successful interventions have been bottom-up interventions that focus on job crafting and mindfulness. Attainment of career success can also be based on employees' various resource management behaviors and attitudes to optimizing their careers [40].

The present study also suggests that the benefits of work engagement may expand outside organizations and concern trade unions, policymakers, and societies in general. For example, in the UK, the Secretary of State for Business ordered an in-depth examination of employee engagement and a report on its potential benefits for companies, organizations, and individual employees that resulted in the "Engaging for Success" report [75]. Similarly, in Finland, the government launched a Working Life 2020 program, which strived to make Finnish work life the best in Europe by 2020. Boosting work engagement was one of its key goals [76]. We hope that the present study will inspire policymakers everywhere to pay closer attention to the potentials of building work engagement and more resourceful workplaces that can help meet the demands of present work life and support sustainability.

### 4.3. Strengths and Limitations

Among the strengths of this study are its large, representative sample of Finnish employees, its use of a longitudinal design and IV analysis to detect causal relationships, and its avoidance of common method bias by linking work engagement with register-based career-related distal outcomes that have not previously been studied in work engagement literature.

However, the study also has limitations. First, although it is based on a nationally representative sample, it only represents one country and its employees. Legislation for pensions and unemployment, as well as wage practices, vary between countries. More research using robust register outcomes in different countries and cultures are needed in the future. Second, the follow-up period of two years may be considered rather short for such distal outcomes. The positive labor market consequences of being engaged at work might be stronger if the follow-up period was longer. We could not extend the follow-up period due to project funding. Third, although we followed employees over a two-year period and used the IV approach, the study design was not a full panel, which would have enabled us to also test reversed and reciprocal relationships. For instance, although our results lent strong support to work engagement predicting increases in wages, pay level could also be a predictor of motivation [77]. We aimed to solve this simultaneous causality issue by using the IV approach.

Fourth, we only studied direct relationships and not the mechanisms linking work engagement with the outcomes, as suggested by the JD-R model. It could be, for instance, that engaged employees build personal and job resources in the long term, which lead to better performance, and in turn, to better wages and occupational positions. In the future, longitudinal research on the various mediators between work engagement and career-related outcomes would be interesting. Perhaps even more importantly, sustainable careers have been broadly defined as "sequences of career experiences reflected through a variety of patterns of continuity over time, thereby crossing several social spaces, characterized by individual agency, herewith providing meaning to the individual" [42] (p. 7). In our study, we were only able to test one sequence of work engagement leading to other indicators of sustainable careers. It is obvious that future research should focus more on the role of work engagement in longer career experiences and processes.

### 5. Conclusions

In this two-year prospective study using a nationally representative sample of Finnish employees, IV estimations, and sustainable careers and COR frameworks, we analyzed the effects of work engagement on future career-related labor market outcomes. We found that work engagement positively impacted employees' working careers in terms of income level and was also positively related to occupational prospects. Work engagement also boosted the ability and willingness to continue working until retirement age, as indicated by a reduced risk of having to retire prematurely because of disabilities, and protected against becoming unemployed during the follow-up period. Thus, the results suggest that the benefits of work engagement may extend beyond those related to job performance and well-being and could concern several important career-related outcomes. As work engagement contains both strong well-being and motivational aspects, it is likely to lead to sustainably healthy and productive working careers.

**Author Contributions:** Conceptualization: J.J.H., P.R., and I.Y.; methodology: P.R., I.Y., and J.J.H.; formal analysis: I.Y.; writing original draft: J.J.H.; editing revision: J.J.H., P.R., and I.Y.; funding acquisition: P.R. and J.J.H. All authors have read and agreed to the published version of the manuscript.

**Funding:** This research was funded by SWiPE research consortium 303667, which is funded by the Strategic Research Council of the Academy of Finland.

**Informed Consent Statement:** This study used data collected by Statistics Finland, which ensured that informed consent was obtained from all individuals involved in the data collection.

**Data Availability Statement:** Researchers should contact Statistics Finland's research services and complete an application for a license to use the data: https://www.tilastokeskus.fi/tup/mikroaineistot/hakumenettely_en.html.

**Conflicts of Interest:** The authors declare no conflict of interest. The funders had no role in the design of the study; in the collection, analyses, or interpretation of data; in the writing of the manuscript; or in the decision to publish the results.

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
