# Peer review of "The Impact of Work Engagement on Future Occupational Rankings, Wages, Unemployment, and Disability Pensions—A Register-Based Study of a Representative Sample of Finnish Employees"

_sustainability, doi:10.3390/su13041626_

Round 1

Reviewer 1 Report

This study is original and interesting. The statistical approach is adequate and robust. The introduction is well structured and review the appropriate literature.

I have only minor suggestions:

  1. In the abstract you may give more indication about the sample size and the results observed.
  2. It would be interesting to develop slightly more explicitly in the introduction how you link the COR and the JD-R frameworks.
  3. In the presentation of the tables and of the statistics be careful to apply more strictly APA rules. "SD", "F" should be in italic; for values that are in-between -1 and 1, the "0" before the "." can be omitted, etc.
  4. It sas difficult to understand why authors have to estimate the wage of 2015.
  5. The sub-section "the present study" was a little bit redundant with the some other parts of the introduction.

Author Response

This study is original and interesting. The statistical approach is adequate and robust. The introduction is well structured and review the appropriate literature.

We thank the Reviewer for these positive comments and have addressed the concerns in the revised paper as follows.

I have only minor suggestions:

  1. In the abstract you may give more indication about the sample size and the results observed.

We added information concerning sample size to the Abstract: “We used nationally representative survey data (N=4876; response rate 68.7%)”. We also added in a sentence describing the results “Even after controlling for outcomes at baseline and several covariates, e.g. health (… we found that work engagement positively predicted future wages and the probability of rising in occupational rankings, and negatively predicted future unemployment and disability pensions)”. As the word limit for the abstract is 200 words and we have now used exactly the word limit it was difficult to add more detailed findings.

  1. It would be interesting to develop slightly more explicitly in the introduction how you link the COR and the JD-R frameworks.

We strengthened the link between COR theory and the JD-R model by adding the following sentence to the Introduction in which we discuss COR theory, work engagement and career outcomes (lines 205-208): “Therefore, as suggested by COR theory, the consequences of the motivational process driven by job resources and work engagement in the JD-R model can be expected to go beyond immediate outcomes such as performance and commitment, and cover career-related outcomes.”

  1. In the presentation of the tables and of the statistics be careful to apply more strictly APA rules. "SD", "F" should be in italic; for values that are in-between -1 and 1, the "0" before the "." can be omitted, etc.

We thank the Reviewer for noticing these careless mistakes and have now corrected them.

  1. It was difficult to understand why authors have to estimate the wage of 2015.

We estimated the wage level both at baseline and at Time 2 so that we could test whether work engagement at baseline predicts wage at Time 2 even after controlling for the baseline wage level.

  1. The sub-section "the present study" was a little bit redundant with the some other parts of the introduction.

We agree with the Reviewer and have now removed the paragraph describing the IV approach from this section and integrated it with what was already written earlier in the Introduction. In addition, we have now clearly separated the hypotheses from the text in which they are developed (lines 261-270).

We also like to inform that we slightly modified the long title and removed the word “follow-up” as reading the rest of the title it was evident anyway. We have now re-titled the paper as: The impact of work engagement on future occupational rankings, wages, unemployment, and disability pensions – A register-based study of a representative sample of Finnish employees. In addition, we slightly simplified the earlier Table 1 (correlations) as the earlier version had two different figures depending on the labor marker status at T2 (employed or not employed).

Reviewer 2 Report

Using the Conservation of Resources (COR) theory and a sustainable careers framework, this study aimed to investigate whether work engagement predicts register-based outcomes of wages, moves in occupational rankings, unemployment, and disability pension.
The article deals with a topic of interest to the journal and is well organized with an excellent methodological foundation.
I report below some considerations that I hope will reinforce the study.
Introduction. The authors are invited to expand the construct of a sustainable career, referring to recent suggestions related to decent work.
Besides, authors are invited to describe the work context related to the sample of Finnish employees that the authors collect to give more specific information about the context.
The authors should revise the section on the research objectives, reducing the bibliographic references that should be included in the preceding paragraphs and pointing the focus to the study's hypotheses.
In the results and data analysis report, the hypotheses that are being attempted to be answered.

Author Response

Using the Conservation of Resources (COR) theory and a sustainable careers framework, this study aimed to investigate whether work engagement predicts register-based outcomes of wages, moves in occupational rankings, unemployment, and disability pension.
The article deals with a topic of interest to the journal and is well organized with an excellent methodological foundation.
I report below some considerations that I hope will reinforce the study.

We thank the Reviewer for these positive and constructive comments.

Introduction. The authors are invited to expand the construct of a sustainable career, referring to recent suggestions related to decent work.

We thank the reviewer for this insightful and important suggestion. Indeed, we think that some similarities with decent work and sustainable careers models have not been discussed in the literature thus far. We have now added the following paragraph. However, as several conceptual and theoretical frameworks are already discussed in the Introduction, and adding one more could confuse the reader, we inserted this paragraph into the Discussion in Theoretical implications section. This is where we discuss another related, widely known concept, the happy-productive worker hypothesis. Adding decent work here seemed to make it flow perfectly well with the other topics in this section (lines 585-599):

Another concept that shares clear similarities with sustainable careers is decent work. Originally proposed as a central goal of the International Labour Organization [67], this concept has four main values: freedom, equity, security, and human dignity; and six key elements: opportunities for work, freedom of choice of employment, productive work, equity, security, and dignity [68], which all fit with the sustainable careers model. More recently, decent work has been conceptualized in different ways, for example, as consisting of five components: physically and interpersonally safe working conditions, access to health care, adequate compensation, hours that allow for free time and rest, and organizational values that complement family and social values [69]. In addition, a recent study [70] used seven dimensions of quality of work (physical environment, work intensity, working time quality, social environment, skills and discretion, prospects, and earnings) to illustrate decent working conditions, which were also associated with work engagement. These conceptualizations of decent work can be considered determinants of sustainable careers. Evidently, future research could benefit from building bridges between these two umbrella concepts.”

Besides, authors are invited to describe the work context related to the sample of Finnish employees that the authors collect to give more specific information about the context.

We have added the following paragraph to describe in more detail the respondents and the context during baseline of the present study (lines 287-292): “In Finland, women exceeded the number of men among wage earners during the data collection at baseline. The respondents were wage earners representing multitude of industries, such as agriculture, forestry, fishing, manufacturing, transport, storage, communication, public administration, social and health care, education, construction and other service sectors. Particularly number of employees working in industry has decreased from 29% in 1977 to 14% in 2013”

The authors should revise the section on the research objectives, reducing the bibliographic references that should be included in the preceding paragraphs and pointing the focus to the study's hypotheses.
In the results and data analysis report, the hypotheses that are being attempted to be answered.

We thank the reviewer for this suggestion. Reviewer 1 also suggested reducing redundancy in the Present study section. We have now removed the paragraph describing the IV approach from this section and integrated it with what was already written earlier in the Introduction. In addition, the hypotheses are now clearly separated from the text in which they are developed (lines 261-270). In addition, in the Results we now clarify whether the results supported our hypotheses.

 A bilingual language consultant has also checked the revised manuscript.

 We also like to inform that we slightly modified the long title and removed the word “follow-up” as reading the rest of the title it was evident anyway. We have now re-titled the paper as: The impact of work engagement on future occupational rankings, wages, unemployment, and disability pensions – A register-based study of a representative sample of Finnish employees. In addition, we slightly simplified the earlier Table 1 (correlations) as the earlier version had two different figures depending on the labor marker status at T2 (employed or not employed).

Reviewer 3 Report

Well written and properly analyzed.

Author Response

Well written and properly analyzed.

We thank the reviewer for this very positive comment.

A bilingual language consultant has also checked the revised manuscript.

 We also like to inform that we slightly modified the title and removed the word “follow-up” as reading the rest of the title it was evident anyway. We have now re-titled the paper as: The impact of work engagement on future occupational rankings, wages, unemployment, and disability pensions – A register-based study of a representative sample of Finnish employees. In addition, we slightly simplified the earlier Table 1 (correlations) as the earlier version had two different figures depending on the labor marker status at T2 (employed or not employed).